# Self-Perturbed Anomaly-Aware Graph Dynamics for Multivariate Time-Series Anomaly Detection

**Jinyu Cai**
Institute of Data Science
National University of Singapore
jinyucai@nus.edu.sg

**Yuan Xie**[*]
School of Computing
National University of Singapore
xieyuan_sss@outlook.com

**Glynnis Lim**
Institute of Data Science
National University of Singapore
glynnis@nus.edu.sg

**Yifang Yin**
Institute for Infocomm Research
A*STAR, Singapore
yin_yifang@i2r.a-star.edu.sg

**Roger Zimmermann**
School of Computing
National University of Singapore
dcsrz@nus.edu.sg

**See-Kiong Ng**
Institute of Data Science
National University of Singapore
seekiong@nus.edu.sg

## Abstract

Detecting anomalies in multivariate time-series data is an essential task across various domains, yet there are unresolved challenges such as (1) severe class imbalance between normal and anomalous data due to rare anomaly availability in the real world; (2) limited adaptability of the static graph-based methods to dynamically changing inter-variable correlations; and (3) neglect of subtle anomalies due to overfitting to normal patterns in reconstruction-based methods. To tackle these issues, we propose Self-Perturbed Anomaly-Aware Graph Dynamics (SPAGD), a framework for time-series anomaly detection. SPAGD employs a self-perturbation module that generates self-perturbed time series from the reconstruction process of normal ones, which provide auxiliary signals to alleviate class imbalance during training. Concurrently, an anomaly-aware graph construction module is proposed to dynamically adjust the graph structure by leveraging the reconstruction residuals of self-perturbed time series, thereby emphasizing the inter-variable disruptions induced by anomalous candidates. A unified spatio-temporal anomaly detection module then integrates both spatial and temporal convolutions to train a classifier that distinguishes normal time series from the auxiliary self-perturbed samples. Extensive experiments across multiple benchmark datasets demonstrate the effectiveness of SPAGD compared to state-of-the-art baselines.

## 1 Introduction

Time-series anomaly detection (TSAD) [Blázquez-García *et al.*, 2021; Zamanzadeh Darban *et al.*, 2024] is an essential machine learning task with significant influence in various domains, such as cybersecurity, industrial systems, healthcare, and finance [Veeravalli *et al.*, 2017; Cook *et al.*, 2019; Schmidl *et al.*, 2022; Zhang *et al.*, 2024b; Qi *et al.*, 2025; Fang *et al.*, 2025]. It aims to identify patterns that deviate from expected behavior over a period of time, and to alert to potential faults,

---

[*]Corresponding Author

39th Conference on Neural Information Processing Systems (NeurIPS 2025).

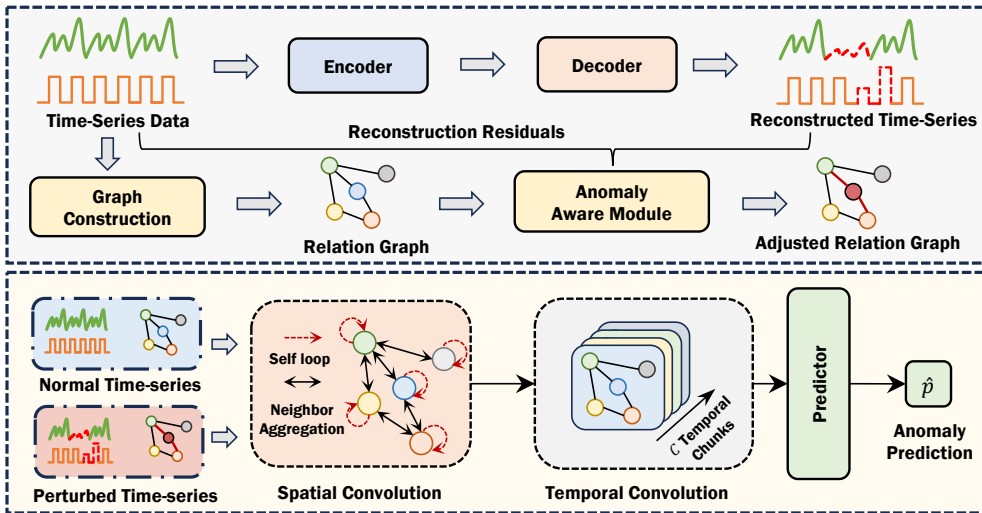

Figure 1: An illustration of the proposed SPAGD framework, which contains three main components: (1) Self-perturbation time-series generation; (2) Anomaly-aware graph construction; and (3) Spatio-temporal modeling for anomaly detection.

intrusions, or critical events. Multivariate time-series anomaly detection, in particular, presents a significant challenge due to the complex spatial and temporal inter-variable dependencies among multiple time-series data.

Over the past decades, TSAD has witnessed significant progress. Early methods are mainly based on proximity [Breunig *et al.*, 2000; Qi and Chen, 2022], linear model [Wold *et al.*, 1987; Schölkopf *et al.*, 2001], clustering [Chen *et al.*, 2022b; Liu *et al.*, 2022; Chen *et al.*, 2022a; Liang *et al.*, 2025; Fu *et al.*, 2025] or outlier ensembles [Liu *et al.*, 2008]. These methods rely on strong assumptions and often struggle with large-scale or high-dimensional data. In recent years, deep learning has emerged as a powerful alternative [Zamanzadeh Darban *et al.*, 2024], with various types of approaches being proposed. For example, the reconstruction-based methods [Audibert *et al.*, 2020; Yang *et al.*, 2023; Wang *et al.*, 2023] employ deep neural network architectures [Baldi, 2012; Goodfellow *et al.*, 2020] to learn latent representations of normal behavior. Anomalies are detected as instances exhibiting significant reconstruction errors, reflecting their deviation from learned norms. Concurrently, forecast-based methods [Tuli *et al.*, 2022; Xu *et al.*, 2022; Wu *et al.*, 2023a] have leveraged sequence modeling techniques [Medsker *et al.*, 2001; Vaswani *et al.*, 2017; Fang *et al.*, 2023] to capture temporal dependencies and predict future values, where anomalies are flagged when observed values diverge from predictions. Although these approaches exhibited effectiveness, they generally treat each variable or time stamp independently, neglecting the rich inter-variable dependencies in multivariate time series. Therefore, recent endeavors [Zhao *et al.*, 2020; Deng and Hooi, 2021] have explored the potential of graph neural networks (GNNs) [Kipf and Welling, 2017; Liu *et al.*, 2023; Wu *et al.*, 2023b; Cai *et al.*, 2024a, 2025b; Liu *et al.*, 2025b] to explicitly model inter-variable dependencies, where variables (*e.g.*, sensors) are represented as nodes in a graph, with edges encoding spatial relationships. Advanced methods [Zheng *et al.*, 2023; Chen *et al.*, 2024] further incorporate GNNs with temporal convolution to capture temporal dynamics for detecting anomalies.

Although these advanced methods provide a promising solution for detecting anomalies in multivariate time-series data, several key challenges still persist:

1. Time-series anomalies are rare or even unavailable in real-world scenarios. The severe imbalance between normal and anomalous time series [Zamanzadeh Darban *et al.*, 2024] could bias the model towards normal patterns during training, thus significantly weakening its generalizability.

2. Existing graph-based methods [Deng and Hooi, 2021; Zheng *et al.*, 2023] typically rely on a static graph construction strategy to model spatial dependencies of time-series data, yet these dependencies may significantly alter due to dynamic anomaly-induced distortion.

3. The "anomaly reconstruction" problem [Audibert *et al.*, 2020; Song *et al.*, 2023] indicates that certain anomalies may inadvertently be well-reconstructed in reconstruction-based methods due to the overfit towards normal patterns, thereby overlooking anomalies with low reconstruction error.

To address these challenges, we propose a new TSAD framework named Self-Perturbed Anomaly-aware Graph Dynamics (SPAGD). SPAGD introduces a self-perturbation module that leverages inherent deviations during the process of reconstructing normal time series to generate self-perturbed time series. These generated time-series data subsequently serve as the auxiliary signal for training the anomaly detection model, where the model is exposed to diverse potential anomalous patterns for alleviating the imbalance problem. Simultaneously, an anomaly-aware graph construction scheme is introduced to learn dynamic inter-variable correlations derived from the evolving reconstruction residuals of self-perturbed time series. Then, we developed a spatio-temporal anomaly detection module to distinguish the self-perturbed time series from normal ones, which effectively integrates both spatial dependencies and temporal dynamics to yield holistic representations for TSAD. SPAGD is trained in an end-to-end manner for the mutual improvement of all components during training. Extensive experiments compared to state-of-the-art TSAD baselines demonstrate the superiority of the proposed SPAGD method. We summarize the main contributions of this paper as follows:

- We introduce a self-perturbation module that generates diverse auxiliary anomalous time series through the evolving reconstruction process of the normal time-series data, which alleviates the severe imbalance problem without relying on any external anomalous data.
- We propose an anomaly-aware graph construction module that dynamically adjusts the graph structure based on the reconstruction residuals of the self-perturbed time series, thus reflecting the changing inter-variable correlations that traditional static strategies fail to capture.
- We build an anomaly detection framework that trains a classifier by exploiting both spatial and temporal dependencies to distinguish normal time series from auxiliary self-perturbed ones, which jumps out of the reconstruction-based framework and mitigates the anomaly reconstruction problem.

## 2 Related Works

Multivariate TSAD [Blázquez-García *et al.*, 2021; Zamanzadeh Darban *et al.*, 2024; Zhang *et al.*, 2024a] is a complex and challenging research area, with a variety of methods emerging in recent years. The sequence-centric models primarily leverage the temporal dynamics of time series, such as Anomaly Transformer [Xu *et al.*, 2022] and TranAD [Tuli *et al.*, 2022], which utilize a Transformer architecture to capture long-term dependencies, while TimesNet [Wu *et al.*, 2023a] enhances detection accuracy through multi-scale temporal decomposition. These methods excel at modeling temporal patterns within individual channels, however, they typically process each variable channel independently, overlooking inter-variable correlations. In recent years, graph-based models have been widely explored to address this limitation, where a graph is constructed to represent the relationships between multiple variables, with GNNs [Cai *et al.*, 2024b; Liu *et al.*, 2024; Cai *et al.*, 2025a] employed for anomaly detection. For instance, GDN [Deng and Hooi, 2021] pre-constructs a similarity graph to help predict anomalies, and GRELEN [Zhang *et al.*, 2022] integrates graph relational learning to improve feature extraction. Moreover, recent approaches [Zheng *et al.*, 2023; Ding *et al.*, 2023] have proposed learning comprehensive representations by incorporating both spatial and temporal dependencies. However, their reliance on fixed graph structures renders them less effective in scenarios where time-series relationships evolve dynamically. Although a few studies have investigated dynamic graph learning [Jin *et al.*, 2024], the graph construction in them is generally decoupled from the anomaly detection optimization process.

The proposed SPAGD method exhibits differences from existing TSAD methods in the following aspects: (1) Rather than relying on random imposed noise [Wang *et al.*, 2023] or traditional augmentation techniques [Yang *et al.*, 2023], SPAGD employs a self-perturbation mechanism to generate auxiliary self-perturbed time-series data, where the perturbations are adaptive and closely tied to anomaly detection. (2) Unlike static graph-based methods [Deng and Hooi, 2021], SPAGD adapts to dynamic changes in inter-variable correlations caused by evolving anomalous candidates. (3) SPAGD leverages the generated auxiliary time series data to train a classifier, which is more flexible and does not require a hand-crafted anomaly-scoring function in reconstruction-based approaches [Song *et al.*, 2023]. (4) SPAGD unifies all components in an end-to-end framework, so that auxiliary time series generation and anomaly detection mutually improve each other.

## 3 Methodology

### 3.1 Problem Formulation

In multivariate time-series anomaly detection, we consider a time series $\mathbf{X} = [\mathbf{x}_1, \mathbf{x}_2, \ldots, \mathbf{x}_T]$ collected across $T$ discrete time steps. Each feature vector $\mathbf{x}_t \in \mathbb{R}^d$ denotes multidimensional observations at timestamp $t$, where $d > 1$ denotes the feature dimensions of the multivariate time-series data. Typically, a time-series dataset $\mathcal{D} = \{\mathbf{X}_1, \mathbf{X}_2, \ldots, \mathbf{X}_N\}$ with $N$ samples is constructed by applying a sliding window of length $T$ to sample a long time series collected from different sources, *e.g.*, industrial sensors. The dataset $\mathcal{D}$ serves as the training foundation for building an anomaly detection model that generalizes to an unknown test dataset $\mathcal{D}_{\text{test}}$. Our goal is to train an anomaly detection model $\mathcal{F}_\Theta : \mathbb{R}^{d \times T} \to \{0, 1\}$ parameterized by $\Theta$ based on $\mathcal{D}$, where the model is able to predict the anomaly state $\hat{y} \in \{0, 1\}$ for each test time sequence $\hat{\mathbf{X}}_i \in \mathcal{D}_{\text{test}}$. Here, $\hat{y} = 1$ and $\hat{y} = 0$ denote anomalous and normal states, respectively.

Despite the abundance of normal time-series data in the real world, current TSAD methods still face several fundamental challenges, such as the **severe data imbalance**, **static spatio-temporal correlation modeling**, and **anomaly reconstruction problem** (refer to Section 1). To tackle these challenges, we propose SPAGD, an end-to-end TSAD framework composed of: (i) self-perturbation time-series generation to alleviate data imbalance; (ii) graph construction to model dynamical inter-variable correlations; and (iii) a spatio-temporal anomaly detection module to mitigate the anomaly reconstruction problem. We will detail each component of SPAGD in the following subsections.

### 3.2 Time-Series Generation via Self-Perturbation

To alleviate the severe data imbalance inherent in multivariate time-series anomaly detection, we propose a self-perturbation mechanism that harnesses the intrinsic imperfections of reconstruction models to generate pseudo-anomalous samples. Our approach is motivated by two key observations. First, during early training stages, the reconstruction model tends to produce systematic errors when replicating the input $\mathbf{X} \in \mathbb{R}^{d \times T}$ due to its limited representational capacity. We argue that these reconstruction errors, which manifest as deviations from the normal data, can serve as effective proxies for genuine anomalies. Second, as the reconstruction model is progressively trained, the magnitude of these errors diminishes, exposing the anomaly detection model to a continuum of deviations, *i.e.*, from large, obvious discrepancies to subtle differences. By incorporating these generated auxiliary samples into the training process as supervised signals, the anomaly detection model is able to progressively refine its decision boundary by identifying a broader spectrum of potential anomalous patterns.

Formally, given an input time series $\mathbf{X}$, we employ a Transformer-based model for reconstruction:

$$\tilde{\mathbf{X}} = \text{Tran}_{\text{d}}\Big(\text{Tran}_{\text{e}}(\mathbf{X}; \Theta_{\text{e}}); \Theta_{\text{d}}\Big), \tag{1}$$

where $\tilde{\mathbf{X}} \in \mathbb{R}^{d \times T}$ denotes the reconstructed time series, and $\text{Tran}_{\text{e}}(\cdot)$ and $\text{Tran}_{\text{d}}(\cdot)$ are the encoder and decoder networks parameterized by $\Theta_{\text{e}}$ and $\Theta_{\text{d}}$, respectively. The reconstruction model is trained on the dataset $\mathcal{D} = \{\mathbf{X}_i\}_{i=1}^N$ by minimizing:

$$\mathcal{L}_{\text{sp}} = \sum_{i=1}^N \Big\| \mathbf{X}_i - \text{Tran}_{\text{d}}\Big(\text{Tran}_{\text{e}}(\mathbf{X}_i; \Theta_{\text{e}}); \Theta_{\text{d}}\Big) \Big\|_F^2. \tag{2}$$

Initially, $\tilde{\mathbf{X}}$ deviates substantially from $\mathbf{X}$, and these deviations are treated as auxiliary anomalous patterns for training the subsequent anomaly detection model. As training progresses and reconstruction quality improves, $\tilde{\mathbf{X}}$ will gradually converge towards $\mathbf{X}$, providing the anomaly detection model with increasingly subtle deviation signals. Particularly, this progressive refinement can dynamically update the quality of pseudo-anomalous samples throughout the training process, which ensures that the anomaly detection model does not overfit to specific types of anomalies but instead generalizes to a broader range of potential anomalies.

### 3.3 Anomaly-Aware Graph Construction

Existing graph-based TSAD approaches [Deng and Hooi, 2021; Zheng *et al.*, 2023] encode each variable (*e.g.*, sensor) as a node ($V$) and pairwise affinities as edges ($E$), allowing GNNs to exploit

cross-variable dependencies via graph $G = (V, E)$, which is unavailable to purely sequence-centric models. However, these methods typically rely on static similarity measures (*e.g.*, cosine similarity) to construct the graph, and keep it fixed throughout training and inference. A static topology can accurately reflect *normal* correlations, yet it fails to capture dynamic disruptions induced by self-perturbed time-series data. To overcome this limitation, we propose an anomaly-aware graph construction (AAGC) mechanism, which can dynamically adjust the graph structure based on the reconstruction residuals of the self-perturbed time series, thereby ensuring that the influence of potential anomalies on the spatial relationships can be emphasized.

**Initial Graph Construction**  Given a normal time-series $\mathbf{X} \in \mathbb{R}^{d \times T}$ from a dataset $\mathcal{D} = \{\mathbf{X}_1, \mathbf{X}_2, \ldots, \mathbf{X}_N\}$, we first construct an adjacency matrix $\mathbf{A}$ to capture baseline inter-variable correlations. For variables $i$ and $j$, we compute pairwise cosine similarity as follows:

$$S_{ij} = \text{Sigmoid} \left( \frac{\langle \mathbf{X}_{i,:}, \mathbf{X}_{j,:} \rangle}{\|\mathbf{X}_{i,:}\| \cdot \|\mathbf{X}_{j,:}\|} \right), \tag{3}$$

where $\mathbf{X}_{i,:} \in \mathbb{R}^T$ represents the time-series for variable $i$, and $\text{Sigmoid}(\cdot)$ is the *Sigmoid* function mapping similarity scores to $[0, 1]$, indicating connection strength. To ensure sparsity and focus on the most informative relationships, we retain only the top-$K$ neighbors for each variable:

$$\mathbf{A}_{ij} = \begin{cases} S_{ij}, & \text{if } j \in \text{top-}K \text{ neighbors of } i, \\ 0, & \text{otherwise.} \end{cases} \tag{4}$$

This step ensures the graph remains sparse, reducing computational complexity while preserving the most relevant inter-variable correlations. The initial graph $\mathbf{A}$ serves as a static representation of interactions between normal multivariate time series.

**Dynamic Adjustment with Reconstruction Residuals**  While static graphs are suitable for capturing inter-variable correlations between normal time-series pairs, they cannot reflect the transient changes (*e.g.*, sensor failures or external disturbances) in dynamic time series. To adapt to dynamic anomalous patterns introduced via generated self-perturbed time series, we construct a dynamically adjusted graph $\tilde{\mathbf{A}}$ to reflect the influence of anomalous candidates on the learned graph structure. To achieve this, we first compute the node-specific reconstruction residual score $r_i$ for each variable:

$$r_i = \frac{1}{T} \sum_{t=1}^{T} |\mathbf{X}_{i,t} - \tilde{\mathbf{X}}_{i,t}|, \tag{5}$$

where $\mathbf{X}_{i,t}$ and $\tilde{\mathbf{X}}_{i,t}$ are the input and reconstructed values for time series $i$ at time $t$, respectively. The reconstruction residual $r_i$ quantifies the anomalous degree of the $i$-th variable within the self-perturbed samples, with higher values indicating greater deviation from normal behavior. Next, we identify the top-$m\%$ variables ranked by their residual scores $r$ as anomalous candidates. The affinity matrix for the self-perturbed time-series is then dynamically adjusted to emphasize these variables:

$$\tilde{S}_{ij} = S_{ij} + \mathbb{I}(i \in \mathcal{M})\phi(r_i) + \mathbb{I}(j \in \mathcal{M})\phi(r_j), \tag{6}$$

where $\phi(r_i) = \frac{1}{1+e^{-r_i}}$ normalizes the anomaly score to a bounded interval $[0, 1]$, and $\mathbb{I}(\cdot)$ is the indicator function. Particularly, for $i = j$, we add $\phi(r_i)$ only once. Following a similar procedure in Eq. (4), we can obtain the adjusted adjacency $\tilde{\mathbf{A}}_{ij}$. Note that we symmetrize the dynamically enhanced similarity before sparsification to ensure numerical stability and a well-posed enhanced structure. This adjustment adaptively boosts the connection strength between any two nodes where at least one is an anomalous candidate. Essentially, compared to static graph construction, the proposed AAGC strategy leverages the node-specific reconstruction residuals $r_i$ to dynamically emphasize connections involving anomalous candidates ($\mathcal{M}$), thereby adapting the graph structure to reflect anomaly-induced changes in inter-variable correlations.

### 3.4 Spatio-Temporal Modeling for TSAD

With the auxiliary time-series dataset $\tilde{\mathcal{D}} = \{\tilde{\mathbf{X}}_1, \tilde{\mathbf{X}}_2, \ldots, \tilde{\mathbf{X}}_N\}$ generated via self-perturbation (Section 3.2) and learned graph structures $\mathbf{A}$ and $\tilde{\mathbf{A}}$ (Section 3.3), we propose a spatio-temporal anomaly detection module designed to effectively capture both instantaneous variable interactions and their temporal dynamics for TSAD. Specifically, the module leverages a unified spatio-temporal representation learning framework and trains a classifier that distinguishes between normal and self-perturbed (auxiliary anomalous) time-series data.

**Dual-Graph Spatial Message Propagation**    To effectively model spatial inter-variable dependencies, we employ a graph attention network (GAT) layer. Let $\mathbf{H}^{(l)}$ and $\tilde{\mathbf{H}}^{(l)}$ be the matrices containing node representations for normal and self-perturbed time series at layer $l$. The GAT updates these representations based on learned attention weights. For each node $i$, the attention coefficient $\alpha_{ij}^{(l)}$ regarding neighbor $j$ is computed by:

$$\alpha_{ij}^{(l)} = \frac{\exp(\text{LeakyReLU}(\vec{a}^{(l)\top}[\mathbf{h}_i^{(l)}\mathbf{W}^{(l)}||\mathbf{h}_j^{(l)}\mathbf{W}^{(l)}]^\top))}{\sum_{u \in \mathcal{N}(i)\cup\{i\}} \exp(\text{LeakyReLU}(\vec{a}^{(l)\top}[\mathbf{h}_i^{(l)}\mathbf{W}^{(l)}||\mathbf{h}_u^{(l)}\mathbf{W}^{(l)}]^\top))}, \tag{7}$$

where $\mathbf{W}^{(l)}$ and $\vec{a}^{(l)}$ are shared learnable parameters for layer $l$, $||$ denotes concatenation, and $\mathcal{N}(i)$ is the neighborhood of node $i$ based on graph $\mathbf{A}$. Similarly, $\tilde{\alpha}_{ij}^{(l)}$ is computed using $\tilde{\mathbf{h}}_i^{(l)}$, $\tilde{\mathbf{h}}_j^{(l)}$ (rows of $\tilde{\mathbf{H}}^{(l)}$) and the neighborhood $\tilde{\mathcal{N}}(i)$ derived from $\tilde{\mathbf{A}}$, using the same parameters $\mathbf{W}^{(l)}$ and $\vec{a}^{(l)}$. The GAT layer then aggregates neighbor information using these attention weights to produce the output matrices $\mathbf{H}^{(l+1)}$ and $\tilde{\mathbf{H}}^{(l+1)}$ for the next layer. This operation, applied node-wise for all nodes $i$, can be represented as:

$$\mathbf{H}_{i,:}^{(l+1)} = \sigma\Big(\sum_{j \in \mathcal{N}(i)\cup\{i\}} \alpha_{ij}^{(l)}\mathbf{H}_{j,:}^{(l)}\mathbf{W}^{(l)}\Big), \quad \tilde{\mathbf{H}}_{i,:}^{(l+1)} = \sigma\Big(\sum_{j \in \tilde{\mathcal{N}}(i)\cup\{i\}} \tilde{\alpha}_{ij}^{(l)}\tilde{\mathbf{H}}_{j,:}^{(l)}\mathbf{W}^{(l)}\Big), \tag{8}$$

where $\mathbf{H}_{i,:}^{(l+1)}$ denotes the $i$-th row of $\mathbf{H}^{(l+1)}$, and similarly for $\tilde{\mathbf{H}}_{i,:}^{(l+1)}$. $\sigma(\cdot)$ is the activation function, such as ReLU. Particularly, $\mathbf{H}^{(0)} = \mathbf{X}$ and $\tilde{\mathbf{H}}^{(0)} = \tilde{\mathbf{X}}$. This mechanism ensures the model learns stable normal spatial relationships via $\mathbf{A}$ while adapting to dynamically changing inter-variable correlations through $\tilde{\mathbf{A}}$.

**Chunk-Wise Temporal Dependency Aggregation**    Following the spatial convolution process, we adopt a chunk-wise temporal graph convolution strategy. For the learned spatial feature $\mathbf{H}^{(L)}$ derived from the final spatial convolutional layer, we partition the temporal axis $T$ into $C$ chunks, *i.e.*, $\mathbf{H}^{(L)} = \{\mathbf{H}_1^{(L)}, \mathbf{H}_2^{(L)}, \ldots \mathbf{H}_C^{(L)}\}$. Similarly, we can apply the same operation to partition the spatial representation $\tilde{\mathbf{H}}^{(L)} = \{\tilde{\mathbf{H}}_1^{(L)}, \tilde{\mathbf{H}}_2^{(L)}, \ldots, \tilde{\mathbf{H}}_C^{(L)}\}$ for the self-perturbed time series. Subsequently, these spatial features are separately processed via a temporal convolution network $\mathcal{T}(\cdot)$ to capture the temporal dependencies:

$$\mathcal{Z}_c = \mathcal{T}(\mathbf{H}_c^{(L)}; \Theta_{\text{Tem}}), \quad \tilde{\mathcal{Z}}_c = \mathcal{T}(\tilde{\mathbf{H}}_c^{(L)}; \Theta_{\text{Tem}}), \tag{9}$$

where $c$ denotes the $c$-th chunk of spatial features, $\Theta_{\text{Tem}}$ denotes the parameters of the temporal convolution network. $\mathcal{Z}_c$ and $\tilde{\mathcal{Z}}_c$ are the learned temporal representations for the $c$-th chunk of normal time series and self-perturbed time series, respectively. We then horizontally concatenate the features learned from each chunk to form comprehensive chunk-wise temporal representations:

$$\mathcal{Z}_T = [\mathcal{Z}_1, \mathcal{Z}_2, \ldots, \mathcal{Z}_C], \quad \tilde{\mathcal{Z}}_T = [\tilde{\mathcal{Z}}_1, \tilde{\mathcal{Z}}_2, \ldots, \tilde{\mathcal{Z}}_C]. \tag{10}$$

**End-to-End Anomaly Detection**    The final spatio-temporal features $\mathcal{Z}_T$ and $\tilde{\mathcal{Z}}_T$ are then vertically stacked as $\mathcal{Z}_{\text{stack}} = [\mathcal{Z}_T; \tilde{\mathcal{Z}}_T]$ and passed through a predictor $\mathcal{P}(\cdot)$, which outputs the final prediction for both normal and self-perturbed time series:

$$\hat{\mathbf{p}} = \mathcal{P}(\mathcal{Z}_{\text{stack}}; \Theta_{\mathcal{P}}), \tag{11}$$

where $\hat{\mathbf{p}} \in [0, 1]^{2N}$ denotes the predicted anomaly probability for the normal and self-perturbed time series, and $\Theta_{\mathcal{P}}$ comprises the network parameters of the predictor. Finally, we train the entire model in an end-to-end manner by jointly optimizing the self-perturbation loss $\mathcal{L}_{\text{sp}}$ (Eq. (2)) and the following anomaly detection loss:

$$\mathcal{L}_{\text{ad}} = -\frac{1}{2N}\sum_{i=1}^{2N}\Big[y_i\log(\hat{p}_i) + (1 - y_i)\log(1 - \hat{p}_i)\Big]. \tag{12}$$

For training the model, we simply set the label for the normal and self-perturbed time series to 0 and 1, respectively. We summarize the overall objective function of the proposed SPAGD framework as:

$$\mathcal{L} = \mathcal{L}_{\text{sp}} + \beta\,\mathcal{L}_{\text{ad}}, \tag{13}$$

where the hyperparameter $\beta$ controls the trade-off between the reconstruction and anomaly detection losses. Particularly, the self-perturbation module is trained with randomly initialized weights, and we explicitly avoid pre-training because the evolution of the reconstruction model is crucial to our proposed learning paradigm. This end-to-end design framework ensures mutual enhancement across the auxiliary time-series generation and spatio-temporal anomaly detection. In the inference stage, we use the output of the trained anomaly detector as the anomaly scores for evaluation. We also provide a detailed algorithm description in the **Appendix C**.

## 4 Experiment

### 4.1 Experimental Configuration

**Datasets** In this paper, we evaluate the proposed SPAGD method on three public time-series datasets collected from different real-world scenarios, including (1) Secure Water Treatment (SWaT) [Mathur and Tippenhauer, 2016], (2) Soil Moisture Active Passive (SMAP) [Hundman *et al.*, 2018], and (3) Mars Science Laboratory (MSL) [Hundman *et al.*, 2018]. Table 1 summarizes the main attributes of these datasets, and we also detail the information of each dataset in the **Appendix A**.

Table 1: Statistics of the three time-series datasets. "AR" denotes anomaly-point ratio in the test split.

| Dataset | Domain | # Channels (Dimensions) | # Train samples (Normal) | # Test samples (Mixed) | AR (%) |
|---------|--------|-------------------------|--------------------------|------------------------|--------|
| SWaT | Water treatment ICS | 51 | 496,800 | 449,919 | 11.98 |
| SMAP | Spacecraft telemetry | 25 | 135,183 | 427,617 | 13.13 |
| MSL | Spacecraft telemetry | 55 | 58,317 | 73,729 | 10.72 |

**Experimental Settings** For a fair evaluation, all baseline methods were trained and tested using identical data splits for each dataset. For SPAGD, we employed a Transformer-based reconstruction model comprising 8 attention heads and 3 encoder layers for self-perturbation learning. During the spatio-temporal modeling stage, each time series sample first undergoes spatial graph convolution that contains 2 graph attention network (GAT) [Veličković *et al.*, 2018] layers with a latent dimension of 256. The learned spatial features were subsequently partitioned into 5 equal-length segments for temporal convolution processing. Aggregated spatio-temporal features were then passed through a predictor consisting of two fully-connected layers to produce final anomaly scores. For other experimental settings and training details, please refer to the **Appendix B**.

**Baselines** We compared SPAGD with extensive state-of-the-art baselines, including: $k$-NN [Ramaswamy *et al.*, 2000], OCSVM [Schölkopf *et al.*, 2001], LOF [He *et al.*, 2003], IForest [Liu *et al.*, 2008], Deep-SVDD [Ruff *et al.*, 2018], COPOD [Li *et al.*, 2020], USAD [Audibert *et al.*, 2020], GDN [Deng and Hooi, 2021], TcnED [Garg *et al.*, 2021], TranAD [Tuli *et al.*, 2022], Anomaly-Trans [Xu *et al.*, 2022], NCAD [Carmona *et al.*, 2022], Deep IF [Xu *et al.*, 2023], TimesNet [Wu *et al.*, 2023a], DCdetector [Yang *et al.*, 2023], COUTA [Xu *et al.*, 2024]. Note that we ensured a fair comparison by reproducing the performance of each baseline using the publicly available codes under the same experiment setting and following the default settings provided in the related papers.

**Evaluation Metrics** To comprehensively evaluate the anomaly detection performance of SPAGD and other baselines, we employ three widely recognized evaluation metrics, including Area Under the ROC Curve (AUC), Area Under the Precision-Recall Curve (AUPRC), and F1-score (F1). We want to specifically highlight that we **_did not_** use the point-adjustment strategy in our evaluation due to its overestimation of model performance [Kim *et al.*, 2022; Liu and Paparrizos, 2024].

### 4.2 Comparison with State-of-the-Art TSAD methods

We comprehensively evaluated SPAGD against extensive state-of-the-art TSAD baselines. Table 2 summarizes the experimental results across multiple benchmark datasets. The key insights are: (1) We can observe that SPAGD showed competitive performance in all three metrics across all datasets. As shown in Table 2, SPAGD achieved AUCs of 86.30% (SWaT), 62.38% (SMAP), and

Table 2: Anomaly detection performance of SPAGD and baseline methods in terms of AUC, AUPRC, and F1 (in %). Note that the best two results are marked in **bold** and underline, respectively.

| Model | SWaT | | | SMAP | | | MSL | | | Avg. |
|---|---|---|---|---|---|---|---|---|---|---|
| | AUC | AUPRC | F1 | AUC | AUPRC | F1 | AUC | AUPRC | F1 | |
| $k$-NN [Ramaswamy *et al.*, 2000] | 77.30 | 68.28 | 71.95 | 39.73 | 11.48 | 25.08 | 59.32 | 17.55 | 32.46 | 44.79 |
| OCSVM [Schölkopf *et al.*, 2001] | 76.06 | 67.27 | 71.79 | 39.87 | 11.46 | 25.08 | 59.33 | 17.58 | 32.61 | 44.56 |
| LOF [He *et al.*, 2003] | 73.33 | 45.23 | 53.18 | 42.94 | 12.35 | 25.40 | 56.37 | 20.06 | 27.61 | 39.61 |
| IForest [Liu *et al.*, 2008] | 78.74 | 66.87 | 66.24 | 39.78 | 11.21 | 25.08 | 56.91 | 16.85 | 29.59 | 43.47 |
| Deep-SVDD [Ruff *et al.*, 2018] | 82.55 | 73.49 | 74.88 | 52.00 | 13.87 | 25.06 | 56.65 | 12.23 | 21.36 | 45.78 |
| COPOD [Li *et al.*, 2020] | 81.66 | 71.57 | 70.85 | 40.04 | 12.02 | 25.08 | 60.56 | 18.17 | **32.77** | 45.85 |
| USAD [Audibert *et al.*, 2020] | 79.67 | 70.24 | 72.60 | 39.51 | 11.41 | 22.69 | 61.67 | 12.86 | 27.12 | 44.19 |
| GDN [Deng and Hooi, 2021] | 81.55 | 71.33 | 75.23 | 58.09 | 16.97 | 19.24 | 49.73 | 12.13 | 22.68 | 45.22 |
| TcnED [Garg *et al.*, 2021] | 82.38 | 72.38 | 76.26 | 53.57 | 12.03 | 19.65 | 48.54 | 12.80 | 23.23 | 44.53 |
| TranAD [Tuli *et al.*, 2022] | 81.85 | 71.62 | 76.45 | 56.44 | 15.24 | 26.51 | 42.97 | 10.03 | 19.35 | 44.49 |
| AnomalyTrans [Xu *et al.*, 2022] | 79.77 | 62.28 | 73.03 | 40.03 | 11.39 | 22.78 | 52.19 | 11.33 | 19.25 | 41.33 |
| NCAD [Carmona *et al.*, 2022] | 19.00 | 8.77 | 21.76 | 39.67 | 11.69 | 22.68 | 59.40 | 14.13 | 21.98 | 24.34 |
| Deep IF [Xu *et al.*, 2023] | 80.77 | 70.67 | 72.97 | **60.89** | 17.33 | **28.48** | 55.97 | 10.98 | 23.09 | 46.79 |
| TimesNet [Wu *et al.*, 2023a] | 32.45 | 14.21 | 21.80 | 39.90 | 11.36 | 22.69 | 55.06 | 11.79 | 21.26 | 25.61 |
| DCdetector [Yang *et al.*, 2023] | 49.68 | 12.20 | 21.97 | 49.79 | 12.76 | 22.70 | 50.10 | 10.55 | 19.09 | 27.64 |
| COUTA [Xu *et al.*, 2024] | 82.95 | 74.78 | 78.68 | 47.20 | 12.56 | 22.69 | 52.17 | 11.84 | 19.08 | 44.66 |
| SPAGD | **86.30** | **77.20** | **78.77** | 62.38 | **18.15** | 27.32 | **66.50** | **21.45** | 30.89 | **52.11** |

66.50% (MSL), outperforming the runner-up baselines, *i.e.*, COUTA, DeepIF, and USAD by 3.35%, 1.49%, and 4.83%, respectively. This observation suggests that the auxiliary signals provided by the generated self-perturbed time series during training ensure the stable performance of SPAGD achieved on different datasets. (2) SPAGD also outperformed reconstruction-based approaches, such as TranAD and AnomalyTrans. This significant performance improvement can be attributed to the designed spatio-temporal anomaly detection module, which leverages the self-perturbed samples to train a classifier instead of reconstruction-based strategies, which helped address the "anomaly reconstruction" problem. (3) Compared to graph-based methods such as GDN, SPAGD demonstrated superior performance. This comparison highlights the significance of the AAGC strategy in SPAGD, which leverages reconstruction residuals induced by self-perturbed time series to dynamically adjust the graph structure, thereby enabling the model to adapt to changing inter-variable correlations that static graph-based methods cannot.

## 4.3 Parameter Analysis

We perform a parameter sensitivity analysis to evaluate the influence of several critical hyperparameters on the performance of SPAGD. Figure 2 illustrates the impact of window size and trade-off parameter $\beta$ on the performance. The experimental results shown in Figure 2(a) revealed that smaller or excessively large window sizes degrade performance to a certain extent, whereas window sizes in the range of $[80, 100]$ consistently yield optimal performance. This is due to the fact that too short a window is not conducive to capturing temporal dependencies, while too long a window leads to difficulties in long-term dependencies learning. Additionally, Figure 2(b) presents the performance under different $\beta$ values, which control the contributions of self-perturbation and anomaly detection losses during training. We can observe a steady performance improvement as $\beta$ increases from $\beta = 10^{-3}$ towards $\beta = 10^{-2}$. Though beyond this point, all metrics slightly decrease, the model maintains relatively stable performance under varying $\beta$ in general. This suggests that overemphasizing anomaly detection loss can lead to overlooking the learning of informative self-perturbed samples for training, which highlights the importance of balancing informative reconstruction and discriminative anomaly detection in practice.

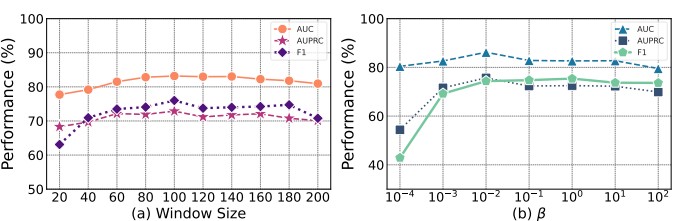

Figure 2: Anomaly detection performance under different window sizes and $\beta$ values on SWaT. Note that we vary the window size and $\beta$ value in a wide range of $[20, 200]$ and $[10^{-4}, 10^2]$, respectively.

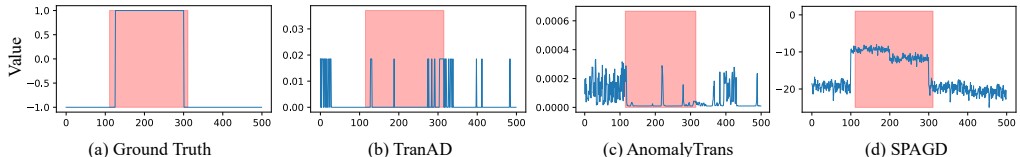

| (a) Ground Truth | (b) TranAD | (c) AnomalyTrans | (d) SPAGD |

Figure 3: Anomaly score visualization on the MSL dataset. We randomly sampled data of length 500 and compared SPAGD with several baselines. The red area indicates the ground truth anomaly.

Table 3: Ablation study results on the three datasets. The best results are marked in **bold**.

| Dataset | SWaT | | | SMAP | | | MSL | | | Avg. |
|---|---|---|---|---|---|---|---|---|---|---|
| Metric | AUC | AUPRC | F1 | AUC | AUPRC | F1 | AUC | AUPRC | F1 | |
| **w/o SP** | 80.30 | 70.34 | 74.04 | 51.31 | 12.49 | 13.25 | 61.37 | 18.75 | 10.54 | 43.60 |
| **SP w/ Recon** | 76.82 | 61.95 | 62.11 | 46.19 | 11.35 | 22.68 | 51.38 | 14.39 | 19.07 | 40.66 |
| **w/o AAGC (Spatial)** | 81.51 | 70.07 | 74.20 | 57.86 | 15.72 | 15.79 | 62.63 | 18.95 | 15.68 | 45.82 |
| **w/o AAGC (Temporal)** | 83.16 | 72.18 | 75.11 | 60.60 | 17.12 | 16.33 | 59.19 | 13.08 | 14.04 | 45.65 |
| **w/o AAGC (Both)** | 80.54 | 70.59 | 74.28 | 55.34 | 16.01 | 23.84 | 54.74 | 12.27 | 19.78 | 45.27 |
| **w/o JT** | 82.97 | 72.01 | 73.83 | 38.60 | 9.98 | 12.79 | 52.09 | 11.45 | 11.21 | 40.55 |
| SPAGD | **86.30** | **77.20** | **78.77** | **62.38** | **18.15** | **27.32** | **66.50** | **21.45** | **30.89** | **52.11** |

## 4.4 Anomaly Score Visualization

We provide a visualization of anomaly scores for an intuitive detection capability comparison. Figure 3 presents the visualization results on MSL, where we compare SPAGD with several baseline methods such as TranAD and AnomalyTrans. We can observe that the anomaly scores of SPAGD are more distinguishable, and the predictions are closely aligned with the ground truth anomaly in Figure 3(a). Other baseline methods exhibit higher detection uncertainty, characterized by frequent false alarms or overlooked anomalies. In particular, we can observe that reconstruction-based methods, such as TranAD, exhibit low reconstruction errors in anomalous events, which further validates the aforementioned "anomaly reconstruction" problem. In contrast, SPAGD is capable of detecting anomalies at these locations. These comparisons visually demonstrate the superior practical detection capability and reliability of SPAGD in real-world scenarios.

## 4.5 Ablation Study

We conducted ablation experiments to evaluate the contribution of each component in the proposed SPAGD method by comparing the SPAGD model against the following variants that shared the same backbones and hyperparameter settings:

1. **w/o Self-Perturbed Generation (SP):** This variant removes the self-perturbation module so that no auxiliary signals are available during training. In this case, the model detects anomalies by solely relying on reconstruction errors.

2. **SP w/ Recon:** This variant retains the self-perturbation module, and uses a traditional reconstruction model to reconstruct both the input and generate auxiliary time series. In the testing stage, the anomaly score is defined by the reconstruction error of a test sample.

3. **w/o Anomaly-Aware Strategy (AAGC):** This variant preserves the self-perturbation module but disables the anomaly-aware strategy in graph construction. Note that we build three variants using a static graph instead of the AAGC strategy for (1) spatial, (2) temporal, and (3) both spatial and temporal modeling, respectively.

4. **w/o Joint Training (JT):** This variant employs a two-stage pipeline where the reconstruction model is pre-trained and then generates the self-perturbed time series once to train the spatio-temporal anomaly detector.

Table 3 shows the experimental results, with the following observations. (1) Removing the self-perturbation module generally leads to a performance drop across all datasets, such as AUCs dropping by 6.00% (SWaT), 11.07% (SMAP), and 5.13% (MSL). This is because the self-perturbation module

provides rich auxiliary potential anomalous signals for model training, which mitigates the class imbalance problem and improves the generalizability. (2) The performance of the "SP w/ Recon" variant significantly decreases, which indicates that a purely reconstruction-based objective encourages the model to fit the self-perturbed patterns rather than distinguish them. Consequently, true anomalies receive lower reconstruction errors and are more difficult to detect in the testing stage, which is a classic example of the "anomaly reconstruction" problem. (3) The performance also decreases when we remove the spatial/temporal or both of them from the AAGC strategy. For example, on the MSL dataset, the performance of the variant that disables AAGC in spatial modeling decreases by 3.87% (AUC), 2.50% (AUPRC), and 15.21% (F1), respectively. This can be attributed to the failure of the model to capture the dynamically changing inter-variable correlations, resulting in the anomaly-induced information within self-perturbed time series being modeled inappropriately. (4) We also observed a significant performance decrease in three datasets after replacing the joint training with a two-stage training paradigm. Generated samples cannot be optimized for the anomaly detection task without joint training, and they are too similar to normal time-series data, which presents significant challenges for the model to distinguish normal time series from them. All in all, the ablation study results provided clear and strong evidence of the effectiveness of each component in SPAGD.

### 4.6   More Experimental Analysis

We further provide more experimental analysis in the **Appendix**, such as extra parameter analysis and visualization results, comparison with different backbone networks and different graph construction methods, comparison with the random-perturbation strategy, and point-adjusted results and more evaluation metrics (VUS-AUC/PR) for reference.

## 5   Conclusion

In this paper, we introduced SPAGD, a new TSAD framework to address several inherent challenges of multivariate time-series anomaly detection. By integrating a self-perturbation module, SPAGD generates diverse auxiliary time-series samples, effectively mitigating the class imbalance as well as providing rich auxiliary potential anomalous signals for training. An adaptive anomaly-aware graph construction method is proposed to dynamically adjust inter-variable correlations for evolving self-perturbed time series. We then train an anomaly detector to distinguish the normal time series from the self-perturbed ones by modeling both spatial and temporal dependencies. Through comprehensive experiments on several benchmark datasets compared to state-of-the-art TSAD methods, SPAGD consistently achieved competitive performance. The parameter analysis, visualization, and ablation study further justify the effectiveness of SPAGD. Despite these advances, SPAGD presumes a relatively homogeneous sensing landscape, which may not be applicable under highly heterogeneous streams [Shao *et al.*, 2024; Jia *et al.*, 2024; Liu *et al.*, 2025a] with asynchronous sensors or weak inter-variable dependencies. Future work could explore heterogeneity-aware graph learning and domain-adaptive perturbation strategies to extend SPAGD to address heterogeneous problems.

### Broader Impact Statement

We provide the broader impact of our work from the following two aspects:

- **Positive social impacts:** Our work has significant potential for enhancing the safety and reliability of critical cyber-physical systems. For instance, early anomaly detection can prevent catastrophic equipment failures, ensuring worker safety and preventing environmental damage. In aerospace applications, it can improve mission success and safety by flagging potential faults in spacecraft telemetry.

- **Potential negative risks:** (1) Over-reliance on the system could lead operators to dismiss their own expertise. **Mitigation:** We advocate for deploying SPAGD as a decision-support tool that provides interpretable outputs (e.g., highlighting anomalous sensors and their correlations) rather than as a fully autonomous system. (2) A sophisticated adversary could potentially craft inputs to either evade detection or trigger false alarms. **Mitigation:** This is a critical area for future work, focusing on improving the model's adversarial robustness.

## Acknowledgement

This research is supported by the National University of Singapore, Institute of Data Science. Any opinions, findings and conclusions or recommendations expressed in this material are those of the author(s) and do not reflect the views of the National University of Singapore.

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
