# OpenReview forum: "Self-Perturbed Anomaly-Aware Graph Dynamics for Multivariate Time-Series Anomaly Detection"
_NeurIPS.cc/2025/Conference — NeurIPS 2025 spotlight_

### Official Review · Reviewer_8kcc · 2025-06-23

**Clarity:** 3
**Significance:** 3
**Originality:** 4
**Rating:** 5
**Confidence:** 4

**Summary:**

In this paper, the authors proposed self-perturbed anomaly-aware graph dynamics (SPAGD), a novel TSAD framework that addresses several crucial challenges in multivariate time-series anomaly detection. Specifically, SPAGD employs self-perturbations derived from reconstruction errors to generate auxiliary anomalous time-series data to facilitate the training of detection model. An adaptive anomaly-aware graph construction module is proposed to adjust graph structures according to the dynamic disruptions within time-series data, and a spatio-temporal modelling framework is integrated to achieve anomaly detection. Experiments conducted on well-established benchmark datasets validate that SPAGD significantly surpasses multiple state-of-the-art baseline methods.

**Questions:**

1. Can the authors provide more details on how the Transformer-based reconstruction model was initialized?
2. What is the rationale for selecting $k$-NN for initial graph construction? How would other alternative graph construction strategies affect the performance?
3. What is the computational complexity of the proposed method? Is it comparable to other baseline methods?
4. Are the performance improvements of SPAGD compared to other baselines statistically significant? Can the authors further conduct statistical significance tests to demonstrate this?
5. Can the authors briefly discuss about the social impact of this work?

**Ethical Concerns:**

["NO or VERY MINOR ethics concerns only"]

**Final Justification:**

In the rebuttal, the authors' response has addressed my concerns. I would like to keep my original positive score.

**Limitations:**

Yes

**Paper Formatting Concerns:**

No concerns

**Quality:**

4

**Strengths And Weaknesses:**

**Strengths**
1. The self-perturbation strategy is a novel and promising solution to the scarcity issue of anomalous data in TSAD. By perturbing normal data, the proposed method generates an evolvable set of auxiliary anomalies to enhance the detection capability of the model without requiring labelled anomalies, which is a common bottleneck in the field.
2. The proposed anomaly-aware dynamic graph construction allows the model to adjust the learned graph structure based on real-time reconstruction residuals. This makes it more adaptable to capture shifting dependencies in real-world applications with complex, evolving systems compared to static graph construction methods.
3. The theoretical analysis in the appendix well justifies the rationale of the self-perturbation and graph dynamics components modules.

**Weaknesses**
1. The description of the self-perturbation module lacks sufficient detail on how the Transformer-based reconstruction model is initialized and how the perturbation evolves during training. For instance, the paper does not clarify whether the reconstruction model is pre-trained or trained from scratch alongside other components, which impacts reproducibility.
2. The claims of performance superiority would be more convincing if supported by statistical significance testing. I suggest the authors conduct appropriate statistical tests (e.g., t-test) to validate that the improvements over key baselines are statistically significant.
3. This paper does not justify the rationale for selecting $k$-NN for initial graph construction. Can another alternative strategy be used for graph construction, and how could this impact the model performance? The authors need to elaborate on it.
4. The computational complexity of the proposed methodology is not currently included. It is recommended that the complexity analysis be supplemented with a discussion of its feasibility for practical deployment.
5. The statement of  "no societal impact" in the checklist is oversimplified for a technology with broad potential applications. I encourage the authors to at least give a brief discussion covering both positive societal benefits and potential negative risks.

---

> ### Author Rebuttal · Authors · 2025-07-30
>
> **We thank the reviewer for the constructive comments. Please find our responses to your concerns below.**
>
> **To W1:** Thank you for pointing out this issue. We clarify that the self-perturbation module is trained with **randomly initialized weights**, and we **explicitly avoided pre-training** because the evolution of the reconstruction model is crucial to our proposed learning paradigm. Specifically:
> 1. In the early training stage, the randomly initialized reconstruction network is a poor approximator of the normal data. Consequently, the generated samples with significant deviations can serve as **easy negative** samples for the training of the classifier.
> 2. As training progresses, the self-perturbation module improves by minimizing the self-perturbation loss ($\mathcal{L}_{sp}$), and the classifier is progressively challenged with **hard negative** samples that deviate more subtly from the normal data manifold.
>
> We will include these descriptions in our revised paper to improve the clarity and reproducibility of our method.
>
> **To W2:** To evaluate the statistical significance of the improvements, we performed a Student's t-test analysis (AUC and AUPRC) on SWaT with 5 independent runs. Our SPAGD achieved **85.48\%$\pm$1.15\% AUC** and **75.67\%$\pm$1.64\% AUPRC**, and below are several baseline performance and corresponding results ($p$-value) of the t-Student's t-test:
>
> $\begin{matrix}
> \hline
> \text{Metric}&\text{AUC}&\text{$p$-value}&\text{AUPRC}&\text{$p$-value}\\\\
> \hline
> \text{Deep-SVDD} &82.48\pm0.36&0.0005&72.76\pm0.94&0.0089\\\\
> \text{TcnED} &82.18\pm0.90&0.0010&72.33\pm1.14&0.0057\\\\
> \text{TranAD} &81.77\pm0.46&0.0002&71.86\pm0.58&0.0012\\\\
> \text{Deep lF} &80.63\pm0.12&0.0000&70.24\pm0.78&0.0002\\\\
> \text{COUTA} &81.43\pm0.97&0.0003&72.39\pm1.41&0.0095\\\\
> \hline
> \end{matrix}$
>
> Generally, **the difference is said to be significant if the $p$-value is less than 0.05**. As shown in the table, the $p$-values are consistently below the 0.05 threshold, which demonstrates that the performance improvements achieved by SPAGD are statistically significant. We will add this experiment to our paper.
>
> **To W3:** Our choice of $k$-NN for graph construction was motivated by its widespread use and proven effectiveness in the GNN-based TSAD works, such as GDN [1]. It is a simple and non-parametric approach that robustly captures local data structure while naturally enforcing graph sparsity, which is computationally beneficial.
> > [1] Ailin Deng and Bryan Hooi. Graph neural network-based anomaly detection in multivariate time series. AAAI, 2021.
>
> To empirically validate this choice, we conducted an additional experiment on SWaT comparing $k$-NN with two other alternatives: (1) a **fully-connected** strategy using all pairwise similarities as edge weights) and an **$\epsilon$-neighborhood** strategy connecting nodes with similarity above a threshold $\epsilon$). Below are the performance comparisons between fully-connected, $\epsilon$-neighborhood and $k$-NN, where we set $\epsilon=0.5$:
>
> $\begin{matrix}
> \hline
> \text{Metric}&\text{AUC}&\text{AUPRC}&\text{F1}\\\\
> \hline
> \text{Full-connected} &82.13&71.79&74.45 \\\\
> \text{$\epsilon$-neighborhood} &81.41&72.53&74.35 \\\\
> \text{$k$-NN} &\bf86.30&\bf77.20&\bf78.77\\\\
> \hline
> \end{matrix}$
>
> The results demonstrate that $k$-NN achieves the best performance, because the fully-connected graph likely introduces noise from weak, irrelevant correlations, while the performance of $\epsilon$-neighborhood is highly sensitive to the choice of the threshold $\epsilon$. Overall, the $k$-NN approach provides a more stable and effective balance. We will include this analysis in our paper.
>
> **To W4:** To address this concern, we analyze the time complexity of SPAGD, which contains three main components:
> 1. **Self-Perturbation module**: This module employs a Transformer-based model, where the computational bottleneck is the self-attention mechanism, which has a complexity of $\mathcal{O}(T^2 d)$ for an input sequence of length $T$ and dimension $d$. As the number of layers is a fixed hyperparameter, the complexity for this module is $\mathcal{O}(T^2 d)$.
> 2. **AAGC module**: The complexity of this module is dominated by the initial graph construction, which involves computing pairwise cosine similarity between all $d$ variables over $T$ timesteps. This results in a time complexity of $\mathcal{O}(d^2 T)$. The subsequent dynamic adjustment steps involve calculating node-wise reconstruction residuals ($\mathcal{O}(dT)$) and adjusting the structure of the sparse graph ($\mathcal{O}(d^{2})$), which are less complex.
> 3. **Predictor module:**
> This module consists of spatial convolution and temporal convolution. The spatial convolution operates on a sparse graph with $|E|=dK$ edges (due to top-$k$ neighbor selection). In our model, the spatial convolution is applied across the temporal dimension $T$ and $L$ layers of GNN with $d_{\text{lat}}$ latent dimensions, resulting in complexity of $\mathcal{O}(LT \cdot dk \cdot d_{\text{lat}})$. Since $L$, $k$, and $d_{\text{lat}}$ are fixed, this simplifies to $\mathcal{O}(Td)$. The temporal convolution complexity is also linear in sequence length and channels, which is approximate $\mathcal{O}(Td)$.
>
> Thus, the overall time complexity of SPAGD for processing a single time-series window is approximately $\mathcal{O}(T^2 d + d^2 T)$, as $T$ and $d$ are dominating factors in the model. For practical deployment, the complexity of SPAGD is comparable to other transformer-based and GNN-based TSAD methods, such as TranAD, AnomalyTrans, and TimesNet. Moreover, we also analyzed the model parameter size of SPAGD (**refer to the response of the reviewer cXmT's W4**), which showed that SPAGD is also highly competitive in terms of model parameter sizes and is feasible for application in a wide range of real-world scenarios.
>
> **To W5:** We are grateful that the reviewer encourages discussions on the broader social impacts of our work. We will add a "Broader Impact Statement" to the paper discussing the following points:
> 1. **Positive social impacts:** Our work has significant potential to enhance the safety and reliability of critical cyber-physical systems. For example, early anomaly detection can prevent catastrophic equipment failures, ensuring worker safety and preventing environmental damage. In aerospace applications, it can improve mission success and safety by flagging potential faults in spacecraft telemetry.
> 2. **Potential negative risks:** (1) Over-reliance on the system could lead operators to dismiss their own expertise. *Mitigation:* We advocate for deploying SPAGD as a decision-support tool that provides interpretable outputs (e.g., highlighting anomalous sensors and their correlations) rather than as a fully autonomous system. (2) A sophisticated adversary could potentially craft inputs to either evade detection or trigger false alarms. *Mitigation:* This is a critical area for future work, where we need to focus on improving the adversarial robustness of the model.

---

### Official Review · Reviewer_cXmT · 2025-06-29

**Clarity:** 2
**Significance:** 2
**Originality:** 2
**Rating:** 4
**Confidence:** 3

**Summary:**

This paper presents a for GNN-Transformer-based framework named Self-Perturbed Anomaly- aware Graph Dynamics (SPAGD) for time series anomaly detection problems. The model consists of two components: a Transformer-based autoencoder and a GNN-based spatiotemporal anomaly predictor. The authors first feed the raw sequence into the autoencoder to obtain a reconstructed sequence (referred to as Self-Perturbation in the paper), which is then passed to a GNN that operates on an adjacency matrix constructed from the reconstruction error (referred to as the Anomaly-Aware Graph). Meanwhile, the original sequence is processed by another GNN based on a static adjacency matrix. The outputs of both GNNs are fed into a prediction network to produce the anomaly scores. Notably, all neural networks are trained jointly. A number of experiments were conducted to verify the effectiveness of the method.

**Questions:**

1.	See the "Weaknesses" section.
2.	How are the anomaly detection threshold selection and localization performed by the authors?
3.	What would happen to the performance if the dynamic error graph were not used and the reconstructed sequence were processed with the static graph instead (which is more efficient)?
4.	Given the introduction of GNN to handle spatial relationships, how do the authors view recent time series forecasting works [3] that have found better regression results by not considering the dependencies (using channel-independence)?

References:
[1] Su, Ya, et al. "Robust anomaly detection for multivariate time series through stochastic recurrent neural network."
[2] Abdulaal, Ahmed, Zhuanghua Liu, and Tomer Lancewicki. "Practical approach to asynchronous multivariate time series anomaly detection and localization."
[3] Nie, Yuqi, et al. "A time series is worth 64 words: Long-term forecasting with transformers."

**Ethical Concerns:**

["NO or VERY MINOR ethics concerns only"]

**Final Justification:**

After careful consideration of the rebuttal and discussions with authors, other reviewers, and the Area Chair (AC), I have decided to maintain my initial recommendation for borderline acceptance. Here are the key points that support this decision:
1.What’s Good: The paper propose a new framework for time series anomaly detection, in which each component is carefully designed. Although at a high level, the method looks like a GAN-style discriminator, it is relatively novel to introduce dynamic and static graph neural networks into such tasks. The experimental results are good, and the authors have subsequently supplemented some additional experiments.
2.What’s Weak: The connection between the theory and the method is not tight, and the authors still have not addressed my concerns. There are logical inconsistencies; for instance, the paper clearly aims to reduce the W-distance between two distributions, yet the authors do not acknowledge this as bringing the two distributions closer. Furthermore, the proposed method is rather weakly connected to the problem it is supposed to solve.
In conclusion, while the authors have made some improvements, the core issues identified during the review process persist.

**Paper Formatting Concerns:**

See the "Weaknesses" section.

**Quality:**

2

**Strengths And Weaknesses:**

Strength:
1.	The writing is clear and easy to understand.
2.	In the experimental section of the paper, SPAGD demonstrates superiority over multiple state-of-the-art methods across three datasets and provides ablation study to prove the necessity of each module, showcasing high quality.

Weaknesses:
1.	The main motivation stated in the Introduction is the anomaly-imbalance problem. However, the proposed method does not clearly explain how this issue is addressed, and there is a lack of corresponding experiments. Moreover, I believe that as long as the model effectively learns the distribution or features of normal data (without overgeneralizing), it should be sufficient to detect any abnormal behavior—especially given that the task focuses solely on determining the presence of anomalies, rather than identifying their specific types. Additionally, it is questionable whether the employed encoder-decoder network is capable of generating or mimicking unseen anomalies.
2.	Setting aside some details (i.e., using a separate dynamic graph for the self-perturbated sequence), the overall approach resembles a standard GAN framework: a Transformer-based reconstruction network serving as the generator, a GNN-based anomaly scorer acting as the discriminator, along with an adversarial loss design. Given that GANs have already been widely applied to time series anomaly detection, this framework is not particularly novel. Moreover, I am skeptical about the effectiveness of the dynamic graph (See the Question 3), especially considering its higher computational cost.
3.	The authors provide some theoretical analysis in the appendix, which is appreciated. However, I find it difficult to assess the practical significance of these results. Specifically, Theorem 1 presents an upper bound on the 1-Wasserstein distance between the distributions of reconstructed and anomalous samples, expressed as the sum of their reconstruction errors relative to normal samples. It is unclear how this bound translates to meaningful insights for anomaly detection, especially given that the reconstruction of anomalous samples is not explicitly considered. A similar concern applies to Theorem 2, where the layer-wise error bounds raise the question of how such perturbations actually affect detection performance in practice.
4.	The experimental section appears somewhat weak, as comparisons are conducted on only three datasets. I recommend including additional datasets such as SMD [1] and PSM [2] to enhance the comprehensiveness of the evaluation. Furthermore, a model complexity analysis would be beneficial—since the results of the ablation study may be affected by reduced parameter counts, the same concern applies to model comparisons.

---

> ### Author Rebuttal · Authors · 2025-07-31
>
> **We thank the reviewer for the constructive comments. Please find our responses to your concerns below.**
>
> **To W1:** Below are our clarifications to address your concern:
> - TSAD is inherently an imbalanced task, and TSAD datasets are mostly highly imbalanced due to the natural scarcity of anomalies in real-world scenarios. Our method attempts to mitigate this problem by progressively generating easy-to-hard auxiliary signals (can be viewed as a **curriculum learning** [1]) to learn a more robust decision boundary. The effectiveness of our solution was verified through **Table 2 in the main text and our response to W4**.
> > Yoshua Bengio, et al. Curriculum Learning, ICML, 2009.
> - We agree that an ideal model that perfectly learns normal patterns without overgeneralizing would be sufficient. However, the "anomaly reconstruction" problem identified in our paper prevents the model from reaching this ideal state, where the learning capacity of reconstruction-based models is powerful enough to accurately reconstruct simple anomalies, leading to low reconstruction errors (refer to Figure 3). Our work is motivated by mitigating this specific failure mode.
> - The goal of our SP module is not to explicitly generate unseen anomalies. Instead, it aims to refine the decision boundary with the self-perturbed samples, which act as **hard negatives** that lie close to the manifold of normal data. By training a classifier to distinguish normal samples from these subtle, model-induced deviations, we forced it to learn a tighter decision boundary around the normal data distribution.
>
> **To W2:** We would like to highlight that our framework is not adversarial. In GAN, the generator and discriminator have competing objectives, while in SPAGD, the components are cooperative. The reconstruction network is not trained to fool the classifier. Instead, as the reconstruction improves, it naturally generates more subtle perturbations, which create a form of curriculum learning that progressively refines the decision boundary.
> - We acknowledge that the AAGC module requires additional computational cost. However, we consider it a crucial component, as a static graph built on normal data hardly captures the dynamic changes of inter-variable correlations caused by anomalies. Our ablation study (**Table 3**) provides direct evidence of its effectiveness, where the "w/o AAGC" variant shows a significant performance drop. Furthermore, we analyze the computational cost of each module in SPAGD (**refer to response to Q2**), and show that the computational cost of SPAGD is still taken up by the transformer-based SP module. Therefore, we consider the slight increase in computational cost to be tolerable given the significant performance improvement (**Table 2**).
>
> **To W3:** We agree that connecting theory to practice is important. Below we discuss the practical significance of our theorems.
> - **Theorem 1:** This theorem provides a formal justification for *why* our self-perturbation strategy is effective. It demonstrates that as the reconstruction model improves during training, the distribution discrepancy between the generated self-perturbed samples and true anomalies decreases. In practice, this means that our training curriculum is not arbitrary but progressively generates more meaningful and challenging samples for the classifier to learn from. This ensures that the auxiliary signals are relevant to our task.
> - **Theorem 2:** This theorem addresses the potential concern that our dynamic graph adjustments could lead to unstable behavior. It provides a formal guarantee that the representations learned by the GNN model remain stable and change reasonably in response to the perturbations introduced by the AAGC module. In practice, this means our model is reliable as the dynamic mechanism will not cause exploding gradients or unstable outputs, which is a critical guarantee for any dynamic system.
>
> In summary, Theorem 1 validates the **relevance** of our generative auxiliary data, and Theorem 2 validates the **reliability** of our dynamic architecture.
>
> **To W4:** We conducted an additional experiment to supplement more datasets and also analyzed the model complexity. We specifically supplemented with the SMD dataset and made a comparison to SPAGD, together with several other baseline methods:
>
> $\begin{matrix}
> \hline
> \text{SMD}&\text{AUC}&\text{AUPRC}&\text{F1}\\\\
> \hline
> \text{Deep-SVDD}&57.10&9.43&14.54\\\\
> \text{USAD}&69.44&25.52&28.19\\\\
> \text{TcnED}&61.43&11.22&18.38\\\\
> \text{TranAD}&71.43&22.19&24.42\\\\
> \text{AnomalyTrans}&69.56&12.72&22.7\\\\
> \text{TimesNet}&70.41&14.48&19.61\\\\
> \text{DCdetector}&49.98&4.15&07.98\\\\
> \text{COUTA}&65.98&11.71&18.03\\\\
> \hline
> \text{SPAGD}&\bf76.92 &\bf38.48 &\bf36.72&\\\\
> \hline
> \end{matrix}$
>
> The experimental results show that SPAGD significantly outperforms the latest baselines, which further justifies its effectiveness.
>
> Regarding the model complexity analysis, we analyzed the parameter size of SPAGD and several baseline methods on SWaT:
>
> $\begin{matrix}
> \hline
> &\text{Network Parameter Size ($\times 10^{6}$)}\\\\
> \hline
> \text{USAD}&84.40\\\\
> \text{AnomalyTrans}&4.85\\\\
> \text{TimesNet}&4.70\\\\
> \text{DCdetector}&0.90\\\\
> \hline
> \text{SPAGD}&2.51\\\\
> \hline
> \end{matrix}$
>
> We observed that the network parameter size of SPAGD is comparable to other state-of-the-art baselines, especially transformer-based ones. These results demonstrate that the performance improvements are not due to a larger model size but are tied to our architectural innovations. We hope our responses are helpful in addressing your concerns.
>
> **To Q2:** The AUC and AUPRC metrics are **threshold-independent**. For the F1-score, we follow the standard method of selecting the optimal threshold on the test set. Localization is achieved by comparing the anomaly scores output by the model for each time window to the selected threshold.
>
> **To Q3:** We would like to answer your question from two perspectives:
> - In our ablation study (**Table 3**), the variants "w/o AAGC (Spatial)" and "w/o AAGC (Temporal)" share the same static graph $A$ for the normal and the self-perturbed time series in spatial convolution and temporal convolution, respectively. For these two variants, a clear performance degradation was observed across all datasets. Here, we further added another variant "w/o AAGC", which shares the same static graph during both spatial and temporal convolution. Below are the experimental results:
>
> $\begin{matrix}
> \hline
> &\text{AUC}&\text{AUPRC}&\text{F1}\\\\
> \hline
> \text{w/o AAGC}&80.54&70.59&74.28\\\\
> \text{w/o AAGC (Spatial)} &81.51&70.07&74.20\\\\
> \text{w/o AAGC (Temporal)} &83.16&72.18 &75.11\\\\
> \hline
> \text{SPAGD} & \bf86.30&\bf77.20&\bf78.77\\\\
> \hline
> \end{matrix}$
>
> The performance of the "w/o AAGC" variant further decreases compared to the other two variants and SPAGD. This empirically justifies the effectiveness of using the dynamical anomaly-aware graph for the perturbed sequence, as the static graph learned from normal data is not equipped to model the altered correlations. We will add this additional ablation study in our paper.
> - Regarding efficiency concerns, we supplement an experiment to compare the computation time cost of (1) the transformer-based SP module, (2) the AAGC module (including the initial graph construction and dynamic graph adjustment), and (3) the predictor (classifier) module over 10 epochs on SWaT:
>
> $\begin{matrix}
> \hline
> &\text{SP module}&\text{Initial graph construction}&\text{Dynamic graph adjustment}&\text{Predictor}\\\\
> \hline
> \text{Time cost in second(s)}&2.03&0.85&0.22&0.01\\\\
> \hline
> \end{matrix}$
>
> The time cost of dynamic graph adjustment is actually small compared to the SP module, as we impose **sparsity** during the graph construction, and the initial graph construction is performed only once during model training. In view of the positive performance improvement (**Table 2**), we consider that the slight increase in time cost is tolerable.
>
> **To Q4:** This is a very insightful question about the current state of research. Our view is that the utility of explicitly modeling inter-variable dependencies is highly **task-dependent**.
> - The goal of forecasting is to predict the *continuation of normal behavior*. In scenarios where channels are not strongly correlated or where learning these correlations introduces more noise than signal, channel-independent models can excel.
> - However, the goal of TSAD is to identify **deviations from normal behavior**, which often manifest as **disruptions in the relationships between variables**. For example, a sensor failing might not produce an anomalous value in isolation, but its readings will no longer correlate with its neighbors. A channel-independent model would be difficult to detect such correlation-based anomalies. For the TSAD task, like SWaT in physical systems, inter-variable dependencies are not noise but an important source of information for detecting system faults.
> Therefore, while channel-independence is a powerful paradigm for certain forecasting tasks, we believe that explicitly modeling spatial relationships is also essential for advancing multivariate time-series anomaly detection. The success of numerous graph-based TSAD methods (such as GDN) has provided evidence for this position.

---

> > ### Comment · Reviewer_cXmT · 2025-08-03
> >
> > Dear Authors,
> > Thank you so much for your responses. Your results look promising. However, I still have some points of confusion. I've listed several quick questions below.
> > -    Regarding W1, first, the concept of curriculum learning is not mentioned at all in the entire text. Second, since SP is not generating or simulating anomalies, doesn’t that contradict the challenges and contributions described in your Introduction?
> > -    Regarding Theorem 1, you proved that the final real anomalies converge in distribution to the output of SP — doesn’t this also contradict your clarification in W1?
> > -    Regarding the number of parameters, why is USAD's parameter count so large (several orders of magnitude higher)? It only has one encoder and two decoders, which is similar to your model architecture — one encoder, one decoder, and one predictor.
> > -    Regarding the computation time, could the authors explain why SP takes several seconds to complete? As I understand it, it only performs a single reconstruction of the sample?
> > I would like to stay with my score.

---

> > > ### Author Response · Authors · 2025-08-04
> > > **Response to follow-up questions**
> > >
> > > We thank the reviewer for providing these follow-up questions. We would like to answer your questions as follows:
> > >
> > > **FQ1:** Below are our clarifications for the two aspects.
> > > - Firstly, we used Curriculum Learning in our rebuttal as a concise academic term to describe an intrinsic mechanism already detailed in the paper. Specifically speaking, the training process of SPAGD naturally creates a curriculum: (1) Initially, the generated self-perturbed samples deviate significantly from the normal data, which acts as "easy" auxiliary signals for the predictor. (2) As training progresses, the self-perturbed samples become increasingly subtle and closer to the manifold of normal data, which function as "hard negatives" that challenge the predictor to refine the decision boundary. (3) This progression **from easy to hard** examples is the essence of curriculum learning. We will add this term and a brief explanation to the revised manuscript to better articulate this mechanism.
> > > - Secondly, we would like to clarify that there is **no contradiction** between the goal of the SP module with the challenges and contributions described in the introduction. The key lies in the definition of the "auxiliary anomalous signals" we generate. The SP module does not attempt to simulate specific, real-world external anomalies. Instead, it generates samples that are "anomalous" only with respect to the model's current understanding of normality. These model-induced deviations are **hard negatives** because they lie near the decision boundary. By forcing the predictor to distinguish normal data from these hard negatives, it learns a tighter and more robust decision boundary. This enhanced boundary then improves the detection of true, unseen anomalies, thereby effectively addressing the challenge of data imbalance without needing to explicitly generate realistic anomalies.
> > >
> > > **FQ2:**
> > > Theorem 1 **does not** prove the distributional convergence between self-perturbed samples and anomalies. In **Assumption 1**, we defined a simplified setting: anomaly is modeled as **normal data + bounded noise ($\eta$)** for the mathematical traceability. Under this setting, Theorem 1 proves that the 1-Wasserstein distance between the distribution of our self-perturbed samples ($\mathbb{P}\_{\tilde{X}}$) and anomalies ($\mathbb{P}\_{anom}$) satisfies:
> > > $ W\_{1}(\mathbb{P}\_{\tilde{X}},\mathbb{P}\_{anom})\le\rho\_{\theta}+\rho\_{anom}$.
> > > Our proof demonstrates that this upper bound **tightens** as the model trains, because the expected perturbation magnitude $\rho_{\theta}$ generated in our model decreases during training ($\rho_{anom}$, is a fixed constant). The decreasing $\rho_{\theta}$ implies our self-perturbed samples are progressively closer to the normal data manifold, which generates increasingly challenging "hard negatives" that help the predictor refine the decision boundary. Consequently, the theorem provides theoretical justification for why our self-perturbation mechanism can generate effective auxiliary signals without explicitly simulating external anomalies, which is consistent with our clarification in W1.
> > >
> > > **FQ3:** The difference in parameter size between USAD and our method stems directly from fundamental architectural choices:
> > > - The USAD's architecture is based on autoencoders built with fully-connected (FC) layers. For time-series data, the input is flattened to a vector of size **"window_size" $\times$ "num_channels"**. The parameter count of an FC layer scales quadratically with the size of its input and output features. For SWaT ("window_size=100", "num_channels=51"), the input feature dimension is 5100. USAD model adopts one encoder of **5100[input dim]-2500[input dim/2]-1250[input dim/4]-10000[latent size = window size(100) x latent dim(100)]** and two **symmetric** decoders. This results in a parameter size of about $84.4\times 10^{6}$, which matches the scale of parameter size that we reported.
> > > - In contrast, SPAGD's SP module uses a Transformer architecture. The parameter size of the core self-attention mechanism scales with the embedding dimension, not the sequence length, making it far more parameter-efficient for long sequences. Our spatio-temporal classifier uses GNNs and temporal convolutions, which are also more parameter-efficient than FC networks. That is why we can see that the parameter sizes of other transformer-based methods are on a similar scale to our method.
> > >
> > > **FQ4:** The "2.03" seconds in our table is not the computational time to process a single sample, but the total training time on the whole dataset that the SP module took for 10 rounds (Epochs). As seen in our response to Reviewer 8kcc's W4, we analyzed that the core self-attention mechanism of the transformer within the SP module has a computational complexity of $O(T^2 d)$. Compared to it, other modules are far lighter. These facts make the computational complexity of the SP module inherently higher than other modules.

---

> > > ### Author Response · Authors · 2025-08-09
> > >
> > > Dear Reviewer,
> > >
> > > As the discussion phase for the rebuttal is coming to an end, we would like to thank you for the constructive comments. If you have any further questions, please do not hesitate to reach out to us for the continuation of the discussion.
> > >
> > > Best regards,
> > > Authors

---

### Official Review · Reviewer_pq7y · 2025-07-02

**Clarity:** 4
**Significance:** 3
**Originality:** 3
**Rating:** 4
**Confidence:** 5

**Summary:**

This paper addresses several challenges in multivariate time-series anomaly detection by proposing a novel approach called self-perturbed anomaly-aware graph dynamics (SPAGD). The core contributions include: (1) a self-perturbation module to generate diverse auxiliary anomalous data, mitigating class imbalance without external anomalous data; (2) an anomaly-aware graph construction to dynamically adapt the learned graph structure based on reconstruction residuals; and (3) a spatio-temporal model integrating both spatial and temporal information between time-series data to identify anomaly.

**Questions:**

See weaknesses.

**Ethical Concerns:**

["NO or VERY MINOR ethics concerns only"]

**Final Justification:**

This well-written paper, logically structured with sound positioning, presents a novel "self-perturbation" method via evolving residuals to tackle TSAD data issues, solves "anomaly reconstruction," shows SOTA-surpassing results.

**Limitations:**

Yes, the authors have adequately discussed the limitations of this paper, particularly the assumption of homogeneous sensor data.

**Paper Formatting Concerns:**

I have no formatting concerns to this paper.

**Quality:**

3

**Strengths And Weaknesses:**

**Strengths**

1. The proposed SPAGD framework is novel and well-motivated, which addresses various challenges in time-series anomaly detection, including anomaly scarcity, adaptation to the changing interdependencies between variables, and anomaly reconstruction problems.

2. The paper is well-structured with precise formulation and illustration. Additionally, the theoretical justifications for employing self-perturbations and dynamically adjustable graphs are compelling and insightful.

3. Reproducible code is provided for enhancing both the reproducibility and credibility of this work.

4. Experiments conducted across multiple real-world datasets provide empirical evidence to support the effectiveness of SPAGD. Further experimental analysis in the appendix offers more insights into the proposed framework.

**Weaknesses**

1. While the idea of this work is novel, the quality and diversity of the self-perturbed samples can be further discussed. The core assumption of SPAGD is that reconstruction residuals of a model can serve as ideal proxies for mimicking real anomalies. However, these "anomalies" are inherently tied to the model's own inductive biases. There may exist a potential risk that the model primarily learns to detect its own reconstruction artifacts rather than a general set of real-world anomalies. This could limit its generalization to real anomalies, as they may differ structurally from the generated perturbations.

2. The additive update rule ($\tilde{A}\_{ij} = S\_{ij} + \sigma(r\_{i})$) in the AAGC module is a key design. Can the authors provide more intuition or explanation for it? Have other alternatives been considered, such as multiplicative scaling?

3. The evaluation relies on traditional point-wise metrics like AUC-ROC and F1-Score, which may not fully reflect the practical time-series anomaly detection performance. The authors should further add segment-based metrics such as the Volume Under the Surface (VUS) to provide a more robust assessment.

---

> ### Author Rebuttal · Authors · 2025-07-30
>
> **We thank the reviewer for the constructive comments. Please find our responses to your concerns below.**
>
> **To W1:** We would like to address this concern by clarifying the role of our self-perturbation module and providing empirical evidence against this potential risk:
>
> - The objective of the self-perturbation module is **not** to generate samples that perfectly mimic the real anomalies. Instead, it aims to leverage a set of generated auxiliary signals to refine the decision boundary. By distinguishing normal data from these **model-induced hard negatives** (self-perturbed samples), the model is trained to become more sensitive to the deviation from the learned normal data distribution, thereby enhancing the ability to generalize to unseen anomalies.
>
> - The perturbations are indeed related to the model's inductive bias. However, these generated samples are not static. They evolve dynamically throughout the end-to-end training process, creating a natural and diverse curriculum for training. This process exposes the classifier to a wide spectrum of deviations that range from large perturbations to subtle perturbations, which prevents the model from overfitting to a single type of perturbation.
>
> - The most compelling evidence against this risk is that SPAGD consistently outperformed state-of-the-art baselines across multiple benchmark datasets (**Table 2** in the main text). If SPAGD were only learning to detect its own internal artifacts, it would fail to generalize effectively across these distinct real-world anomalies. The superior performance of SPAGD strongly suggests that the learned decision boundary is indeed robust and generalizable. Furthermore, our results on segment-based VUS metrics (**see the response to W3**) further validate this robust generalization.
>
> **To W2:**  We would like to address your concern from two perspectives:
>
> - The intuition behind the additive rule is to treat the graph affinity as a combination of **static correlation** and **dynamic anomaly-awareness**. $S_{ij}$ represents the baseline correlation between variables $i$ and $j$ under normal conditions. $\sigma(r_i)$ is a score derived from the reconstruction residual of variable $i$. When a variable is highly perturbed (high $r_i$), we strengthen the connection of all its edges. This effectively highlights the subgraph surrounding a potential anomaly, which enforces the message-passing mechanism to focus on these regions.
>
> - We did consider alternatives, such as the multiplicative scaling suggested by the reviewer (e.g., $\tilde{A}\_{ij} = S_{ij} \cdot (1 + \sigma(r\_i))$). However, we found the additive rule to be more effective, as the additive rule generally outperforms the multiplicative rule by $\geq$2\% across all metrics (on SWaT) in our attempts. This is because the effect of the multiplicative rule is conditional on the initial similarity $S_{ij}$. If $S_{ij}$ is near zero, the multiplicative strategy has a very limited effect. This is a significant limitation, as an anomaly could cause a new, strong correlation to emerge between two previously uncorrelated variables. The additive rule can capture this phenomenon by creating emphasis even on connections that were initially weak, making it more flexible for modeling diverse anomalous behaviors.
>
> **To W3:** We conducted additional experiments to evaluate the performance of SPAGD in terms of VUS-ROC and VUS-PR metrics. Below are the experimental results on SWaT, where we compared the performance with several state-of-the-art baseline methods:
>
> $\begin{matrix}
> \hline
> \text{Metric}&\text{VUS-ROC}&\text{VUS-PR}\\\\
> \hline
> \text{USAD}&53.78&32.86\\\\
> \text{TcnED}&62.99&44.30\\\\
> \text{AnomalyTrans}&57.54&36.02\\\\
> \text{Deep IF}&56.48&37.56\\\\
> \text{TimesNet}&44.49&21.74\\\\
> \text{DCdetector}&52.31&15.01\\\\
> \text{COUTA}&77.10&51.86\\\\
> \hline
> \text{SPAGD}&\bf84.93&\bf62.08\\\\
> \hline
> \end{matrix}$
>
> We can observe that SPAGD outperforms the latest baselines on these two metrics, which fully supports the effectiveness of our method. We will include this experiment in our paper.

---

> > ### Comment · Reviewer_pq7y · 2025-08-07
> >
> > Thank you for your responses—my concerns have been largely addressed.

---

> > > ### Author Response · Authors · 2025-08-08
> > >
> > > We sincerely thank the reviewer for acknowledging our work and are glad to see that your concerns have been addressed.
> > > We will incorporate your suggestions into the revision to further strengthen the clarity and experimental evaluations of our paper.

---

### Official Review · Reviewer_gEyj · 2025-07-03

**Clarity:** 3
**Significance:** 3
**Originality:** 3
**Rating:** 5
**Confidence:** 4

**Summary:**

This paper proposed SPAGD for time-series anomaly detection. Specifically, SPAGD introduces the self-perturbation time-series generation, anomaly-aware graph construction, and spatio-temporal anomaly detection modules to correspondingly tackle three key challenges in the field: the imbalance between available normal and anomalous data, the limitation of static graph structures in capturing dynamic correlations, and the "anomaly reconstruction" problem. The entire framework is trained in an end-to-end way. The authors conducted comparative experiments on various benchmark datasets, showing that SPAGD outperforms a wide range of TSAD baselines.

**Questions:**

The authors should focus on addressing my concerns raised in Strengths and Weaknesses, which will affect my final assessment of this paper.

**Ethical Concerns:**

["NO or VERY MINOR ethics concerns only"]

**Final Justification:**

The authors provided a thorough rebuttal that addresses my concerns. In particular, the inclusion of evaluations using more robust VUS metrics and additional experimental analysis significantly strengthens the empirical validations of this paper. In light of these clarifications and improvements, I decide to raise my rating to Accept.

**Limitations:**

yes

**Paper Formatting Concerns:**

Although I understand the main body of the paper is limited, I suggest that the authors include the main claims of the theoretical analysis in the main text, if possible, which will increase the readability of this paper.

**Quality:**

3

**Strengths And Weaknesses:**

Strengths:
1. This paper is well-written and easy to follow. The problem statement is clear, and the technical details are explained logically. The related work section positions SPAGD well within the existing literature.
2. Leveraging the evolving reconstruction residuals as a source of "self-perturbation" to generate training signals is novel and sound. This directly addresses the critical and practical challenge of data scarcity and class imbalance in TSAD without relying on external data or simple noise injections. This paper also identifies the "anomaly reconstruction" problem and proposes a promising solution to address it.
3. The experimental results demonstrated that SPAGD significantly outperforms state-of-the-art TSAD baselines on various benchmark datasets, and the authors also provided the implementation code of SPAGD.

Weaknesses:
1. The paper adopts a "chunk-wise" temporal convolution, which breaks the continuity of the time series. What is the motivation for this design over standard sequential models (like LSTMs) that preserve temporal continuity? Does this approach potentially fail to capture dependencies that span across chunk boundaries?
2. Although the authors have included several commonly used metrics in their evaluation, the TSAD community is increasingly adopting more robust metrics such as VUS-ROC and VUS-PR [1]. To strengthen empirical validation, I suggest the author supplement the evaluation of the proposed method using these two metrics.
3. The hyperparameter $m$, which is the percentage of anomalous candidates, appears critical for the AAGC module, yet its impact is not analyzed. How does it affect the model's performance?
4. The current ablation study is insightful, but the variant "w/o SP" removes both the auxiliary data and the classification objective. Would it be possible to conduct a more fine-grained ablation study to better isolate the contribution of individual components? For example, a variant that uses self-perturbation but with a reconstruction-based objective could help quantify the specific benefit of the classification objective.

[1] Paparrizos et al. Volume under the surface: a new accuracy evaluation measure for time-series anomaly detection, Proceedings of the VLDB Endowment, 2022.

---

> ### Author Rebuttal · Authors · 2025-07-30
>
> **We thank the reviewer for the constructive comments. Please find our responses to your concerns below.**
>
> **To W1:**
> We would like to address your concern via the following two perspectives:
>
> 1. **With regards to motivation:**
> - Sequential models like LSTMs process data recursively over time, which can be computationally intensive and slow for long time-series windows. Temporal convolution, on the other hand, can process sequences in parallel, leading to better training efficiency.
> - The chunk-wise design allows the model to learn fine-grained temporal patterns within each local chunk. The subsequent concatenation and predictor network then learn higher-level relationships between these chunks. This hierarchical approach is effective for capturing both local dynamics and the overall structure of the time window, which has shown success in recent TSAD models such as TimesNet [1].
> > Haixu Wu, et al. Timesnet: Temporal 2D-variation modeling for general time series analysis. ICLR, 2023.
>
> 2. **Regarding the dependencies across chunk boundaries, we agree that this is an important aspect. Our framework mitigates this potential issue in two ways:**
> - The spatial graph convolution is applied to the representation of the entire time series before the chunking operation (**Eq. 7**). This ensures that information is propagated across all time steps via the learned graph, thus preserving a holistic view of the window.
> - The final predictor module takes the concatenated representations from all chunks as input (**Eqs. 9 and 10**). This allows the predictor to explicitly model the relationships and dependencies between the learned chunk features, thereby capturing patterns that span across chunk boundaries.
>
> **To W2:** VUS-ROC and VUS-PR are indeed robust metrics to evaluate TSAD methods. To address your concern, we conducted additional experiments on SWaT to evaluate SPAGD and several key baselines using these two metrics. Below are the experimental results:
>
> $\begin{matrix}
> \hline
> \text{Metric}&\text{VUS-ROC}&\text{VUS-PR}\\\\
> \hline
> \text{USAD}&53.78&32.86\\\\
> \text{TcnED}&62.99&44.30\\\\
> \text{AnomalyTrans}&57.54&36.02\\\\
> \text{Deep IF}&56.48&37.56\\\\
> \text{TimesNet}&44.49&21.74\\\\
> \text{DCdetector}&52.31&15.01\\\\
> \text{COUTA}&77.10&51.86\\\\
> \hline
> \text{SPAGD}&\bf84.93&\bf62.08\\\\
> \hline
> \end{matrix}$
>
> As shown in the table, SPAGD demonstrates superior performance across all datasets under these two challenging metrics. This once again strengthens our claim about the effectiveness of SPAGD. We will include these results (with more datasets) and a detailed discussion in our paper.
>
> **To W3:** We performed a sensitivity analysis of $m$ on the SWaT dataset. Below are the anomaly detection performance under different $m$ in the range of $[10\\%,60\\%]$:
>
> $\begin{matrix}
> \hline
> m&\text{10\\%}&\text{20\\%}&\text{30\\%}&\text{40\\%}&\text{50\\%}&\text{60\\%}&\\\\
> \hline
> \text{AUC}&80.74&83.22&\bf86.30&83.30&83.32&81.58\\\\
> \text{AUPRC}&71.00&72.45&\bf77.20&71.82&73.13&71.39\\\\
> \text{F1}&74.07&74.37&\bf78.77&75.76&74.76&75.48\\\\
> \hline
> \end{matrix}$
>
> The results showed that the performance of SPAGD is robust within a reasonable range of $m$ (e.g., 20\% to 50\%). We can observe that the performance degrades when $m$ is too small (e.g., 10\%), as the AAGC module lacks a strong enough signal to dynamically adjust the graph structure. Conversely, if $m$ is too large (e.g., 60\%), the performance also slightly decreases as an excessive number of normal variables are incorrectly flagged as anomalous candidates, which introduces noise into the graph structure. We will include related experiments and analysis in the paper.
>
>
> **To W4:** As per your suggestion, we designed a new variant, i.e., **SP w/ Recon**: This variant retains the self-perturbation module, and uses a traditional reconstruction model to reconstruct both the input and generate auxiliary time series. In the testing stage, the anomaly score is defined by the reconstruction error of a test sample. The experimental results are shown as follows:
>
> $\begin{matrix}
> \hline
> \text{Metric}&\text{AUC}&\text{AUPRC}&\text{F1}\\\\
> \hline
> \text{SP w/ Recon}&76.82&61.95&62.11\\\\
> \text{SPAGD} & \bf86.30 & \bf77.20 & \bf78.77\\\\
> \hline
> \end{matrix}$
>
> We can observe that the performance of the "SP w/ Recon" variant significantly decreases, which indicates that a purely reconstruction‑based objective encourages the model to fit the self‑perturbed patterns rather than distinguish them. Consequently, true anomalies receive lower reconstruction errors and are more difficult to detect in the testing stage, which is a classic example of the "anomaly reconstruction" problem. This experiment validates the contribution of our discriminative (classification) objective in mitigating the "anomaly reconstruction" problem. We will include these results in the paper.
>
> **To Paper Formatting Concerns:** Thank you for this suggestion. We agree that presenting the key theoretical insights in the main text would make the paper more self-contained, and we will revise the paper accordingly.

---

> > ### Comment · Reviewer_gEyj · 2025-08-03
> >
> > The authors provided a thorough rebuttal that addresses my concerns. In particular, the inclusion of evaluations using more robust VUS metrics and additional experimental analysis significantly strengthens the empirical validations of this paper. In light of these clarifications and improvements, I decide to raise my rating.

---

> > > ### Author Response · Authors · 2025-08-04
> > >
> > > Thank you for your positive feedback and acknowledgement. We are glad that our rebuttal can successfully address your concerns. We will revise and improve our manuscript according to your suggestions.

---

### Note · Authors · 2025-08-14

We sincerely thank the Area Chair and all reviewers for their time and insightful feedback. This rebuttal discussion period has been an invaluable opportunity for us to improve our paper. We are grateful for the constructive suggestions provided. As detailed in our rebuttal, we have supplemented the following:

- **More Robust Metrics:** We supplemented the evaluation using VUS-ROC and VUS-PR, where SPAGD consistently shows superior performance.

- **Additional Dataset:** We supplemented additional experimental validation on the new SMD dataset.

- **Deeper Analysis:** We supplemented new ablation study results and more hyperparameter sensitivity analysis.

- **Statistical Significance Tests:** We conducted Student's t-test to justify the statistical significance of our performance improvements over baseline methods.

- **Complexity analysis:** We analyzed the theoretical complexity of our method and provided empirical comparison in terms of network parameter scale with several baselines.

- **Social Impact Discussion:** We discussed both potential positive and negative social impacts of our method.

We appreciate the genuine engagement from our reviewers, which allowed us to clarify the core contributions of our work. We have also refined our explanation, such as how self-perturbation acts as a curriculum of hard negatives to learn a robust decision boundary, and the practical significance of the theorems. We are thankful that our response was acknowledged by the reviewers and received positive feedback. We hope that these clarifications have resolved your concerns.

---

### Decision · Program_Chairs · 2025-09-17

**Decision:**

Accept (spotlight)

**Comment:**

There is a clear consensus in the PC that this is a strong paper that should be accepted. Congratulations to the authors! I am requesting the author to carefully incorporate the suggestion provided by the review in camera-ready version.